Visual and acoustic components of courtship in the bird-of-paradise genus Astrapia (Aves: Paradisaeidae)

Scholes Edwin es269@cornell.edu edwin.scholes@cornell.edu 1
Gillis Julia M. 2
Laman Timothy G. 3
1 Cornell Lab of Ornithology, Cornell University , Ithaca , NY , United States of America
2 Center for Animal Resources and Education, Cornell University , Ithaca , NY , United States of America
3 Museum of Comparative Zoology, Harvard University , Cambridge , MA , United States of America
Hedrick Ann
Electronic publication date: 2017 Nov 8
Publication date: 2017
Volume: 5
Electronic Location ID: e3987
Received 2017 Jul 20; Accepted 2017 Oct 12
Copyright: ©2017 Scholes et al.
Copyright year: 2017
Copyright holder: Scholes et al.
License: This is an open access article distributed under the terms of the Creative Commons Attribution License, which permits unrestricted use, distribution, reproduction and adaptation in any medium and for any purpose provided that it is properly attributed. For attribution, the original author(s), title, publication source (PeerJ) and either DOI or URL of the article must be cited.
License URL: https://creativecommons.org/licenses/by/4.0/

Keywords: Display behavior, Visual signaling, Acoustic signaling, Video analysis, New Guinea, Courtship phenotype

Funding: The authors received no funding for this work.

==============================
The distinctive and divergent courtship phenotypes of the birds-of-paradise make them an important group for gaining insights into the evolution of sexually selected phenotypic evolution. The genus Astrapia includes five long-tailed species that inhabit New Guinea’s montane forests. The visual and acoustic components of courtship among Astrapia species are very poorly known. In this study, we use audiovisual data from a natural history collection of animal behavior to fill gaps in knowledge about the visual and acoustic components of Astrapia courtship. We report seven distinct male behaviors and two female specific behaviors along with distinct vocalizations and wing-produced sonations for all five species. These results provide the most complete assessment of courtship in the genus Astrapia to date and provide a valuable baseline for future research, including comparative and evolutionary studies among these and other bird-of-paradise species.

Introduction

The birds-of-paradise (Aves: Paradisaeidae) are a well known, sexually selected, radiation of species, celebrated for their bewildering diversity of courtship behaviors and exotic plumages (Frith & Beehler, 1998; Scholes, 2008a; Laman & Scholes, 2012). The approximately 40 species have evolved over roughly 20 million years, with most ornamental phenotypic evolution occurring within the core birds-of-paradise (i.e., all but the basal Manucodia/Lycocorax clade) over the last 15 million years (Irestedt et al., 2009). The diversity of their ornamental traits and the extent of their phenotypic modifications relative to other birds make them an important group for gaining insights into sexually selected phenotypic evolution (Scholes, 2008a; Martin, 2015; Arnold & Houck, 2016).

While they are typical passerine birds in most regards, nearly all bird-of-paradise species have distinctive and divergent courtship phenotypes (Scholes, 2008a). In males, the courtship phenotype is the portion of the phenotype that is perceived and evaluated by females during courtship and in females it is the portion devoted to mate evaluation and selection (Scholes, 2008a). The courtship phenotype of a species includes visual and acoustic components, as well as the morphological and behavioral components that underlie them.

An important part of understanding the evolution of phenotypic diversity the birds-of-paradise therefore requires detailed knowledge of the visual and acoustic components that comprise their courtship phenotypes (Scholes, 2008a; Scholes, 2008b; Scholes, 2008c). Yet for many species, descriptions of courtship phenotypes are only available in the broadest of terms or, for some species, lacking entirely. For the most part descriptions of courtship phenotypes in the birds-of-paradise lack the requisite detail need for comprehensive comparative studies that can yield insights into phenotypic evolution (Scholes, 2008a). For instance, nearly all of species in the genus Astrapia are still lacking even basic descriptions of their courtship phenotypes.

The five species in the genus Astrapia (Viellot, 1816) are medium sized, compact bodied, and unusually long-tailed species, which inhabit higher-elevation montane forests of mainland New Guinea (Mayr & Gilliard, 1952; Gilliard, 1969; Frith & Beehler, 1998; Frith & Frith, 2009; Frith & Frith, 2010; Laman & Scholes, 2012; Beehler & Pratt, 2016). Three species—A. nigra (Gmelin, 1788), A. splendidissima Rothschild, 1895 and A. rothschildi Foerster, 1906—are allopatric while the other two—A. mayeri Stoner, 1939 and A. stephaniae Finsch and Meyer, 1885—overlap with each other at the margins of their respective elevation ranges in a small part of Papua New Guinea’s central highlands (Frith & Beehler, 1998). Molecular phylogenetic data suggest the genus is monophyletic, roughly 6 million years old and forms a sister group with the two species in the genus Paradigalla (Irestedt et al., 2009). Astrapia and Paradigalla are members of a larger clade that includes the other long-tail birds-of-paradise from the genus Epimachus (Irestedt et al., 2009).

Male Astrapia appear mostly black under most lighting conditions, but have highly iridescent greenish-blue heads, an intensely reflective coppery-orange band on the upper breast and either a deep violet (nearly black) or mostly white tail. The three allopatric species—A. nigra, A. splendidissima and A. rothschildi—also have iridescent green lower breasts and bellies and a conspicuous iridescent green or purplish nape/mantle ‘cape’ (Frith & Beehler, 1998; Frith & Frith, 2009; Laman & Scholes, 2012; Pratt & Beehler, 2015). Female plumaged birds are mostly dull blackish brown with lighter barred undersides (Frith & Beehler, 1998; Pratt & Beehler, 2015). Breeding behaviors are not well known, but all species are thought to be polygynous, with promiscuous males that use arboreal display sites, and with females providing all parental care (Frith & Beehler, 1998).

Behavioral components of courtship for species in the genus Astrapia are very poorly known (Healey, 1978; Frith & Beehler, 1998; Frith & Frith, 2009; Frith & Frith, 2010; Laman & Scholes, 2012). There is no inventory of systematically named and described courtship behaviors or vocalizations for species in the genus. Summaries from species accounts in the secondary literature indicate a few simple behaviors including a form of hoping back and forth between branches and an inverted display posture in A. rothschildi (Healey, 1978; Frith & Beehler, 1998; Frith & Frith, 2010).

Here, we use an existing collection of audiovisual recordings from the Macaulay Library, a publicly accessible repository of animal behavior media, to fill in gaps in knowledge about the visual components of the courtship phenotypes for four species in the genus Astrapia and the acoustic components for all five species. Results, while still fragmented, nevertheless provide the first holistic documentation of courtship phenotype diversity in the genus Astrapia. Because results are derived from and documented with permanently vouchered audiovisual data, they can be readily reanalyzed and reinterpreted, which offers a solid foundation for future research, including comparative and evolutionary studies among these and other bird-of-paradise species.

Materials & Methods

Our analysis was performed using previously archived audiovisual recordings available in the Macaulay Library at the Cornell Lab of Ornithology (https://www.macaulaylibrary.org/). The complete dataset contained 301 audiovisual media records collected in the field between the years 1974 and 2011 (with the majority collected by authors ES and TGL between 2001 and 2011). The dataset, which is unequally distributed among species, includes audiovisual media for all five species and has been fully accessible for research through the Macaulay Library’s online catalog since 2012.

It is important to note that while two authors (ES and TGL) were the field collectors for most of media in the dataset, the recordings were collected as part of a family-wide audiovisual documentation effort spanning a 10-year period and were not collected as a part of a focused field study on the genus Astrapia. As is common with studies based on analysis of museum specimens, the dataset does not include some of rich contextual information often included in a focal study where data were collected for the sole purpose behavioral and/or acoustic analysis (e.g., details of forest structure/ecology around display sites, height and location of display perches in the forest canopy, presence and behavior of other individuals not recorded, etc).

Listed as they occur from west to east throughout New Guinea, the dataset includes data from all five species: (1) A. nigra with nine video recordings, one audio recording and six photographs from one subadult male, which was responding to audio playback of its own vocalization; (2) A. splendidissima with 31 video recordings and three audio recordings from an unknown number of adult males; (3) A. mayeri with 86 video recordings, nine audio recordings and six photographs from at least four individual adult males; (4) A. stephaniae with five video recordings, 10 audio recordings from an unknown number of individuals; (5) A. rothschildi with 129 video recordings, one audio recording and five photographs from three individual adult males and at least two females, but possibly more.

Because the initial dataset was not well annotated for behaviors when it was originally compiled, the entire dataset was pre-screened and parsed for media containing evidence of behavioral and acoustic components used in the context of courtship. The parsed media subset was then annotated using appropriate behavioral terms from the Macaulay Library database controlled vocabulary. Only media with courtship-relevant acoustic and visual elements were included in the final analysis. The refined and annotated dataset can be retrieved through the online catalog by performing a taxonomic search for Astrapia and filtering with the behavior terms: perform courtship display, perform visual display, mate, vocalize, call and mechanical sound.

Next, videos from the refined dataset were obtained from the Macaulay Library as archival quality QuickTime movie files (.mov) in the Apple ProRes 422 codec. For playback, we used QuickTime Player and/or Final Cut Pro on Apple iMac, Macbook and desktop computers. Videos were screened again to identify important ethological components and to apply provisional names to them (e.g., “perch-hop”). Videos with diagnosable behavioral units were scrutinized at normal and frame-by-frame playback speeds to identify and describe basic action patterns, associated movements and context. Representative videos were selected as reference vouchers.

As needed, video frames highlighting the important components of the behaviors were extracted and manipulated in Adobe Photoshop. Video frame image manipulations were minor adjustments to improve clarity, including changing levels, brightness, contrast, and de-interlacing interlaced video with the default de-interlacing filter. Image modification was done only to enhance ability to view the frame in instances where video quality was poor for creating figures.

For the acoustic components, archival quality audio files were obtained from the Macaulay Library as 44khz, 16-bit WAVE audio files (.wav) or extracted from video and converted into 44khz, 16-bit .wav files mating the format of the independent audio files. Audio was imported into the sound analysis software package Raven Pro version 1.5 for analysis (Cornell Lab of Ornithology, Ithaca, NY, USA).

Results

For heuristic purposes, results are presented in order of the species with more results to those with less. To the extent possible given the data, descriptions of visual and acoustic components are presented in the order they are hypothesized to occur within the display sequence. However, because sample sizes are small, the hypothesized sequence order should be considered tentative.

Visual components of Astrapia rothschildi—Huon Astrapia

We identified seven distinct courtship and mating behaviors, five male and two female behaviors. The five male behavors are: (1) the perch-hop, which is a series of short hops among branches, (2) the flick-pivot, which is a repeated turning in place from side-to-side while flicking the wings open and shut, (3) the inverted tail-fan, which is an inverted display that features elaborate fanning of the elongated tail feathers, (4) the upright nape-peck, which is a vigorous ritualized pecking at the nape of a female and (5) the post-copulatory tumble, which is a highly unusual behavior that involves the male and female spiraling toward the ground together after copulation. The two female behaviors are: (1) a display interaction behavior, which involves the female actively engaging with the courting male and (2) a wing-flutter, which is a simple solicitation behavior. Each of these behaviors is described in detail below.

Courtship was observed to take place in the forest canopy from specific, small to mid-sized, largely horizontal, branches (perches) from one or several adjacent trees used for display—i.e., display sites. To the extent of the data available, display sites appear fairly open and have multiple perches in close proximity that are routinely used for display, with one perch serving as a primary display perch.

Perch-hop

This display involves a series of quick short hops or flight-hop among several perches (video vouchers: ML 458060, 458224 and 458238). It takes place prior to and upon arrival on the primary display perch as well as after the behaviors described below and therefore is a “bookend” flanking the other displays. While most perch-hops take place before or after the arrival of a female, there are nevertheless times when the hops or flights are used to chase female plumaged birds (both females and sub-adult males) and other adult males. Chasing occurs both to and from the display perches and also from the canopy outside of the display site.

Flick-pivot

This is a simple, but common, introductory behavior in which the male adopts a horizontal posture and repeatedly turns in place from side-to-side while flicking the wings and tail open (video vouchers: ML 456250, 458062 and 458225). It begins from the display perch, with the male’s body held in a horizontal posture, with head up, bill pointing forward and tail hanging below the perch nearly perpendicular to the body (Fig. 1A). The green feathers of the flanks and belly are sleeked against the body and the plush feathering of the breast and throat are fluffed, but not pushed forward toward the bill (Fig. 1B). From this posture, the head and upper body are repeatedly turned from side-to-side as if on a pivot (i.e., from a fixed central point) in a ritualized motion. With each pivot, the body slightly rises and lowers in-place, but the feet don’t leave the perch and the wings and tail are quickly “flicked” i.e., spread open and shut in rapid succession (see voucher videos). The wings are only partially extended from the body. Tail flicking occurs by rapidly spreading and shutting the outer retrices so that the tail broadens and then relaxes. When the wing flicks are done forcefully, it is accompanied by a brief non-vocal shhek sound (see below), which is acoustically similar to the wing-produced sonation emitted in flight (see below).

Figure 1 Flick-pivot of A. rothschildi.

The flick-pivot is initiated from a horizontal posture, with head up, bill forward, green ventral plumage sleeked against the body and tail hanging below the perch (A). The plush feathering of the breast and throat are fluffed, but not pushed forward toward the bill (B). The head and upper body are moved from side-to-side in a ritualized pivot motion as the wings are rapidly spread open and shut, i.e., “flicked”. A variation includes horizontal pointing, which involves lowering the head, neck outstretched with bill pointing forward and tail lifted to be horizontal and inline with the body (C). Image credit/source: (A) Timothy G. Laman/ML 59349941, (B) Timothy G. Laman/ML 456250, and (C) Timothy G. Laman/ML 456246.

Between pivots, the male will occasionally engage in a bout of perch-hopping and hop from one display branch to an adjacent one before resuming flick-pivoting on both. The flick-pivot often includes a 180-degree rotation to face the opposite direction, sometimes hopping back and forth between two perches repeatedly (e.g., ML 456250 and 458238 at time 00:33–00:55, which shows flick-pivoting with two perch-hops with 180-degree rotations). Female plumaged birds are not on display perch during the flick-pivot and this display takes place prior to the higher intensity display that lead to copulation (see below).

A variation of the flick-pivot includes an additional component that we call horizontal pointing (video voucher: ML 456246). It occurs irregularly when first arriving on a display perch or when hoping back-and-forth between branches during flick-pivots with perch-hopping. Horizontal pointing involves the male lowering his head and slightly outstretching the neck with his bill forward and body horizontally rigid while simultaneously lifting the tail to be horizontal and inline with the body/head. The result is that for a brief moment, the bird appears to be “pointing” with its entire body forming a horizontal line from bill to tail-tip (Fig. 1C). Pointing ends with the tail being slowly relaxed and dropped to perpendicular behind the body and typical flick-pivots begun or resumed.

Inverted tail-fan

This high-intensity display is routinely performed to females. It involves the male rotating himself backwards, tail-first, around the axis of the perch to become inverted with his undersides facing skyward (video vouchers: ML 458066 and 458217). It takes place from a horizontal perch, usually the main display perch, and often follows a bout of flick-pivoting. When a female is present, the display begins with the male intently facing the approaching female in a rigid body posture that is usually low and horizontal to the perch, but is sometimes upright (Fig. 2A). The feathers of belly, flanks and lower breast are held sleeked against the body and the plush ornamental breast and throat feathers are erected outward to form a black circular shape with an intense orange fringe that nearly encircles the head (Figs. 2 and 3). The feathers of the throat and chin are pushed forward to extend outward around the bill, which is closed, giving the head a somewhat conical appearance (Figs. 2B–2D). As a female approaches, the male lowers himself backwards, tail-first, until he has rotated around the axis of the perch to become inverted with the intensely iridescent green feathers of the belly and the black underside of the tail are facing skyward (Figs. 2B–2C and 3A). In this position, the male hangs upside-down with his main body axis aligned, head to tail, mostly perpendicular to the branch, but sometimes slightly rotated to one side, the upper body curled slightly upward toward the branch so that the bill is pointed skyward or held close to the perch (Figs. 2C and 3A). While inverted, with tail pointing upward, the male repeatedly moves the tail in an exaggerated rocking motion, which causes the tail to wave up-and-down vertically. A the same time, the male rhythmically fans the tail by expanding the tail feathers outward from the middle pair, which sometimes part to form a gap, and then shutting them (Figs. 2C and 3B). In this position the bird’s entire body forms a crude L-shape or C-shape depending on if the head is back or pointed up toward the female (Figs. 2C, 3A and 3B). While the male is in the inverted tail-fanning posture, he repeatedly lunges, bill first, upward toward the female (or toward the branch) in semi-regular intervals. The upward lunging movement serves to further exaggerate the waving and fanning motion of the tail.

Figure 2 Inverted tail-fan of A. rothschildi.

From an upright posture similar to the flick-pivot with ventral plumage sleeked and plush breast feathers erected (A), the male backs, tail first, to hang below the perch in an inverted posture (B). While inverted, the feathers of the throat are pushed forward around the bill giving the head a conical appearance (C). While inverted, the green ventral plumage and the underside of the tail face skyward and the tail is spread and fanned (C). Plate (C) inset illustrates the general body posture of the bird in the photo. Red arrows and dotted line are in the same positions for both the inset and photo. In the illustration, the tail is more inline with the body and not spread whereas in the photo the tail is being lifted toward the head and spread. The erected breast feathers form a circular disk-like shape with an intense orange fringes that encircles the head (D). Image credit/source: (A) Timothy G. Laman/ML 59343391, (B) Timothy G. Laman/ML 59343381, (C) Timothy G. Laman/ML 59543371, and (D) Timothy G. Laman/ML 59343401.

Figure 3 A. rothschildi inverted tail-fan and upright nape-peck with female interactive observation behavior.

While inverted, with bill pointed toward the female and tail cocked upward to form a crude L-shape, the tail is repeatedly fanned and moved in a side-to-side rocking motion (A). From this position, the male tracks the movement of the female while repeatedly lunging, bill first, toward her (B). When the male is inverted, the female approaches the male in a rigid upright posture from above with tail fanned and held beneath the perch at a right angle to the body axis (A). From the inverted tail-fanning posture, the male begins the upright nape-peck behavior by rotating, head first, around the perch into an erect upright posture to face the female with ventral plumage sleeked and breast feathers still erected around the head (C and D). With tail cocked under the perch and bill pointed at the female, the male repeatedly lunges at the nape of the female causing her to turn away and/or move away from the male (C and D). Image credit/source: (A and B) Edwin Scholes/ML 458213 and (C and D) Edwin Scholes/ML 458080.

Throughout his time in the inverted posture, the male will ‘track’ the movement of the female so that as the female moves along the perch, the male adjusts his position according to hers, tracking her movement to keep his bill pointed toward her. While high-intensity and performed for females, the inverted tail-fan display does not lead to copulation directly (see next).

Upright nape-peck

This high-intensity behavior is a pre-copulatory display. It involves the male returning upright to face the courted female while vigorously and rhythmically lunging and pecking at the females nape (video vouchers: ML 458069, 458080, and 458213). It begins immediately after an extended bout of inverted tail-fanning if the female remains on the perch. The transition from inverted tail-fanning to nape-pecking happens when the female stops directly interacting with the male and turns to face away with wing-fluttering (Fig. 3C). Sometimes, the female will move along the branch away from the male before turning away and wing-fluttering such that the male must follow before proceeding.

From the inverted tail-fanning posture, the male rotates, head-first, around the perch into an erect upright posture with plumage along the belly still tightly sleeked and the breast shield fully fanned into a distinct disk-shape around the upper body (Fig. 3C). The tail is cocked under the perch so that the entire body forms an upright semi-circle from head to tail-tip (Fig. 3C). The bill is pointed directly forward facing the female and tail-fanning continues as the male begins to vigorously lunge at the nape of the female (Figs. 3C and 3D). The nape-pecking motion is an intense and rhythmic lunge forward toward the female followed by a rigid rearing back with a slight pause on the back end. Often, nape-pecking consists of a rapid double or triple forward lunge after the rear pause. Sometimes the pecking motion becomes so intense it appears as if the male is not only pecking, but raking his bill along the neck and back of the female. After several seconds of pecking and lunging toward the female, who is vocalizing with each peck (described below), the male will mount, wings open and flapping, by standing on her back, bill pointed toward her head, and plumes fully expanded.

Post-copulatory tumble

This unusual behavior involves the male standing on the back of the female after copulation and flapping his wings while leaning forward over until the pair tumble, entangled together, off the branch toward the ground (video vouchers: ML 458069 and 458213). It begins immediately following copulation when the male repositions himself to be standing on the back of the female. The male spreads and flaps his wings while leaning forward over the female until the pair drop forward off the branch and tumble, locked together, toward the ground. The falling often involves a dramatic spiraling and twisting downward through the canopy (e.g., ML 458215). Though no video captures how far they fall, it is clear that it is at least several meters and possibly much more.

Of ten mating events documented in the dataset, seven resulted with the pair tumbling entangled together and three resulted in the two becoming separated before the male fell forward so that the female was able to fly off in a separate direction. In a few instances (e.g., ML 45077), the female does not release her grip on the perch and, for a brief moment, both male and female hang under the perch in a flapping mass of wings and tails before tumbling together.

Female display interaction

This behavior involves the courted female interacting with the displaying male by somewhat mirroring the postures and behaviors of the male (video voucher: ML 458079). It takes place while a male is performing the inverted tail-fanning display and the observing female approaches the male in an upright posture from above on the same display perch. As a female gets closer to the male, she adopts a more rigid body posture with the tail fanned and held beneath the perch at a right angle to the axis of the body in a way that loosely mirrors the crude L or C-shape of the male below (Fig. 3A). As the female moves toward the male to observe him at close range, she too often moves toward or away from the male to become partially inverted like the male. Her movements are tracked and countered by the male, which in turn causes her to react and move in order to maintain her position relative to his. After several seconds of engagement and intense up-close observation and interaction with the male, the male begins to move closer and/or lunge toward the female, which often causes the female to turn away and move further away from the male along the perch. If the male remains inverted, the female will often return for another bout of intense interactive observation behavior. If she stays further away or continues to move away, the male will usually right himself and the female will then either transition into giving the wing-fluttering behavior (below) or exit the display perch(s) entirely.

Female wing-flutter

This female behavior involves the female distancing herself slightly from the displaying male while turning away and fluttering her partially spread wings (video voucher: ML 458069). It begins after a prolonged bout of intense observation and intense inverted display by the male (Fig. 3C). Wing-fluttering continues while the male approaches and during the vigorous upright nape-pecking behavior of the male. While wing-fluttering, the female will give a series of nasal “shriek” call (see below) that are timed with the apparent impact of the lunging bill pecks of the male.

Acoustic components of Astrapia rothschildi—Huon Astrapia

We identified two sounds used in the context of courtship, one vocal and one non-vocal (audio voucher: ML 147993 and video vouchers: ML 456104, 456109 and 456255). The most common acoustic component of courtship is a non-vocal sonation produced by the wings during flight and as the wings are flicked open and shut during display, e.g., the flick-pivot display. In flight, the wing sonation has the acoustic quality of a baby rattle and sounds like a repeated, “shhek-shhek-shhek” with each wing beat (Fig. 4A). Each “shhek” note has a very broad frequency range and short duration (Fig. 4A). When emitted during the wing-flicks of pivot display, the sound is either a single “shhek” or a slight double “shhek-shhek” sound.

Figure 4 Spectrograms of Astrapia non-vocal wing sounds.

In all five species, each broad frequency pulse is thought to correspond to a single wing beat. The sustained “shhek-shhek-shek” of a male A. rothschildi (A). The louder rattle-like sounds of A. mayeri, which include visible pauses between bouts of multiple wing beats (B). The sounds of A. nigra, which are similar to A. rothschildi (C). The fainter and more temporally separated rattle-like sounds of A. splendidissima (D). The loud and conspicuous rattle sounds of A. stephaniae, which sound most similar to A. mayeri (E).

The only documented vocalization is a quiet, frog-like throaty, “rawk” sound given by the male while perched upright (Fig. 5A). The “rawk” vocalization typically ranges from one to three rawks emitted in sequence. Notes are spaced approximately one second apart (Fig. 5A). As the male vocalizes the tail swings forward and then back slightly. Due to the length of the tail, this movement is fairly conspicuous.

Figure 5 Spectrograms of Astrapia vocalizations.

Two “rawk” notes of A. rothschildi (A). The whistled two note upward-sweeping vocalization of A. mayeri (B). The single “click” note of A. nigra (C). The three-note “jeer-jeer-ti” vocalization of A. splendidissima (D). The whistled two-note “wee-wheet” of A. stephaniae (E).

Female plumaged birds (and presumed females) were recorded to give a series of loud nasal sounding cries and “shrieks” as the male pecks and grabs at her nape with his bill directly prior to copulation (see pre-copulatory nape-pecking above). The shrieks are reminiscent of nasal sounding notes of a Melidectes honeyeater. No other vocalizations were detected for female plumaged birds.

Visual components of Astrapia mayeri—Ribbon-tailed Astrapia

We identified four distinct courtship and mating behaviors: (1) the perch-hop, which is rapid set of hops from branch to branch, (2) the hunchback-pivot, which is a ritualized turning from side-to-side in a distinctive hunched posture, (3) the upright sleeked posture, which is a ritualized upright body posture with ornamental plumage held tight against the body and (4) the branch-sidle, which is lateral movement along a branch in a horizontal posture. Each behavior is described in detail below.

As no complete sequence was documented, the order of the behaviors described below is tentatively assigned based on available data. Because data come from practice displays performed at what appears to be feeding sites and not established display sites, no generalization about display site location and attributes can be made. Similarly, because none of the documented behaviors were high-intensity versions performed directly to females being courted, female position, behavior and interactivity are lacking.

Perch-hop

A simple behavior involving brief flying hops from branch-to-branch, first in one direction and then back, all in rapid succession (video voucher: ML 465722). In the only documented instance of this behavior, the wing-rattle sound is very prominent (see below). As he lands on the perches that are visible (several are off camera), he maintains in the same body posture as in the pivot display (see below), where the legs are spread, the body is ridged, horizontal and the back appears hunched. From that posture, for the perches visible within the frame, he performs a single pivot (see below) before flying off the branch, which suggests that perch-hopping and pivoting co-occur (i.e., perch-hopping, pivoting, perch-hopping, pivoting, etc). In one instance, perch-hopping behavior is initiated immediately following a pseudo-mating event in which the male performs a “practice” pre-copulatory display (see below), but it seems to be the start of another round of display rather than being continuation of copulation sequence.

Hunchbacked-pivot

This display involves the male pivoting from side-to-side while adopting a distinctive hunchbacked posture (video vouchers: ML 465288, 465289, 465689). It begins with the male standing high on the perch with legs spread and outstretched so that tibias are clearly visible (Fig. 6A). The neck is outstretched and the body is held rigidly horizontal and parallel to the perch in a distinctive hunchbacked posture (Figs. 6A–6D). In the hunchbacked posture, the feathers along the ventral side are held tight to the body and the plush feathers along the back (mantle cape) are erected, which creates the distinctive hunchbacked appearance. The rounded “pompom-like” forehead tuft is pushed forward over the outstretched and forward pointing bill. The iridescent green feathers on the top of the head are positioned to accentuate the brilliance of the green when viewed from head-on, while the plush black feathers from around the neck are pushed forward to create a striking contrasting green and black head pattern (Fig. 6E). Just before the pivot, the tail is lifted so that the black feathers are inline with the bill-body axis and the long, white central feathers dangle in a wide arc behind the bird. From this position, the body is then pivoted rapidly to one side and back. Most pivots involve small lateral hops along the branch, but some are stationary—i.e., the feet stay in same place on the perch. The rapid side-to-side movement causes the long white tail feathers to lift up and be swished up and to the side in an exaggerated, but stereotyped, way (Figs. 6B–6D). During practice or low motivation versions and/or in birds with shorter center tail feathers, the movement is the same but the tail swishing is less pronounced. The position from which females observe or interact with males performing the hunchbacked-pivot is unclear from the available data, but it is likely to be an introductory display like the flick-pivot of A. rothschildi.

Figure 6 Hunchbacked-pivot of A. mayeri.

The male stands high on the perch with body rigidly horizontal, tail hanging, legs outstretched and dorsal side of the body in a distinctive hunchbacked posture (A). The body is rapidly pivoted to one side and back, which causes the long tail feathers to lift up and be swished in an exaggerated way (B–D). While in the hunchbacked posture, the pompom-like forehead tuft is pushed forward over the bill and the green feathers on the top of the head are positioned to accentuate the brilliance of the green when viewed form head-on (E). Note: the second bird in the background of plate E is a hybrid between A. mayeri and A. stephaniae with intermediate features, including the shape and color of the tail and reduced forehead tuft. Image credit/source: (A) Timothy G. Laman/ML 59350331, (B) Timothy G. Laman/ML 59350351, (C) Timothy G. Laman/ML 59350341, (D) Timothy G. Laman/ML 59350321 and (E) Timothy G. Laman/ML 59350371.

Upright sleeked posture

This simple behavior involves the male adopting a highly ritualized upright posture with plumage sleeked (video voucher: ML 465288 and 465692). In the two documented instances, it starts with the bird in a horizontal perching position and suddenly lifting the body upright such that the upper body becomes erect, elongated and perpendicular to the perch (Fig. 7A). While adopting the upright and outstretched posture, the ventral plumage of the upper body is sleeked against the torso, which when viewed from the front, highlights both the reddish feathering of the belly and the iridescent green of the throat and neck. While upright, the tail hangs directly below the bird and is in-line with the body axis. During the display, the bird appears to be looking intently at something in the distance, with the bill pointing and slightly upturned. The behavior in video ML 465692 is incomplete, with recording stopping before the bird moves out of the posture. In both documented instances, the male lowers himself back down and adopts a low horizontal body posture with plumage puffed. This posture looks similar to the one adopted during branch-sidling (described below) and why we tentatively place upright sleeked posture before the branch-sidle. Like the previous display, the position from which females observe or interact with males preforming this behavior is unclear from the available data.

Figure 7 Upright sleeked posture and pre-copulatory nape-peck of A. mayeri.

While looking intently at something distant, the male performs the upright sleeked posture by lifting and outstretching the upper body until it is nearly perpendicular to the perch with plumage sleeked against the torso (A). During the pre-copulatory nape-peck behavior, the male crouches in a hunched posture and then repeatedly lunges, becoming rigid and upright, while pecking with his bill (B and C). In the images above, the male is pecking at a knob on the branch (the “practice female”). Image credit/source: (A) Timothy G. Laman/ML 465692, (B) Timothy G. Laman/ML 465689 and (C) Timothy G. Laman/ML 465689.

Branch-sidle

This enigmatic behavior involves the male adopting a low horizontal posture while performing a series of slight lateral hops along the perch (video voucher: ML 465670). Only one instance of this behavior was documented. It begins from a horizontal hunched position similar to the pivot, but with body low and closer to the perch and more-or-less parallel to the ground. The bill is pointed forward as if peering intently at something just out of view. From this position the male performs a series of slight lateral hops along the perch while maintaining the low horizontal position and bill pointing directly ahead. While hoping, the feet are momentarily lifted off the perch to move sideways, or to sidle, along the branch. In the one documented instance, after sidling in one direction, the male turns his body parallel to the branch and moves the other direction up the branch until he is about a meter on the other side of his initial position and hops in place for several seconds before terminating the display. Like the previous two displays, the position from which females observe or interact with males preforming this behavior is unclear from the available data.

Nape-peck

In a presumed pre-copulatory display similar to that of A. rothschildi, the male performs a vigorous nape-pecking behavior toward a female (video voucher: ML 465689 and 465722). In two documented practice versions without a female present, the male crouches in the hunched posture similar to that of the pivot and then adopts an ridged posture with sleeked ventral plumage (similar to that of the upright sleeked posture, but not as upright) and repeatedly lunges at vertical knob on a branch (i.e., the “practice female”) in a rigid ritualized motion (Figs. 7B and 7C). With each lunge the male pecks at the “female” with his bill. In one of the documented instances, the male copulates with the branch at the end of pecking before suddenly hopping off and performing the branch-hopping display (ML 465722 at time 00:09). In the other instance, after the ritualized pecking, the male immediately begins a bout of hunchbacked-pivoting followed by a perch-hopping departure.

Acoustic components of Astrapia mayeri—Ribbon-tailed Astrapia

We identified two sounds used in the context of courtship, one vocal and one non-vocal (audio voucher ML 163699 and video voucher: ML 465678). Like A. rothschildi, the most prominent acoustic component is the wing-rattle sonation, which is given in flight and when the wings are used while hopping back-and-forth between branches during the branch-hopping display. The wing sound of A. mayeri is louder and has an even more distinct rattle-like quality than in A. rothschildi (Fig. 4B). For the vocalization, the male gives two upward-sweeping notes, which sound like “eert” “weet”. The second note is of higher pitch than the first, but both plaintive and slow (Fig. 5B).

Visual components of Astrapia nigra—Arfak Astrapia

We identified one courtship display behavior, the inverted tail-fan, which is an inverted display that features elaborate fanning of the tail feathers. It was performed as a practice display without a female present. Observations about female position, behavior and interactivity are lacking. As with the data from A. mayeri, since data come from behaviors performed at site not thought to be an established display site, there are no inferences about display site location and other physical display site attributes to be made.

Inverted tail-fan

This behavior strongly resembles the inverted display of A. rothschildi (video voucher: ML 455276). Male plumage is held in much the same position: feathers along the undersides are flattened while feathers around the face and neck are extended forward. As with A. rothschildi, the body is rotated backward to drop below the perch, tail first, with both bill and tail pointing skyward to form a crude C-shape from head to tail-tip (Fig. 8A). When fully rotated backward with the tail pointing upward over the display perch, the bird gives a brief thrust forward and gives the click vocalization. With the upward thrust, the tail drops slightly backward and retrices are spread outward from the central pair in a tail-fanning motion similar to that of A. rothschildi (Fig. 8B).

Figure 8 Inverted tail-fan display of A. nigra.

With feathers along the ventral side flattened and feathers around the face and neck extended forward, the body is rotated backward to drop below the perch, tail first, with both bill and tail pointing skyward (A). With an upward thrusting motion, the tail drops slightly and tail feathers are spread in a tail-fanning motion similar to that described for A. rothschildi (B). Image credit/source: (A) Timothy G. Laman/ML 59352091 and (B) Timothy G. Laman/ML 59352131.

Acoustic components of Astrapia nigra—Arfak Astrapia

Similar to the A. rothschildi and A. mayeri, A. nigra is not particularly vocal. We identified two sounds used in the context of courtship, one vocal and one non-vocal (audio voucher: ML 163723 and video vouchers ML 455272 and 455273). The documented vocalization is a single, short low frequency click, or clock note that sounds similar to human “tongue–clicking” or the sound of two pool balls gently knocking together (Fig. 5C). Before vocalizing, the head is slightly lifted so that the bill points upward and opens slightly, then the head is snapped down sharply as the bill is brought back to the level as the bird produces the click sound.

In sustained flight, the subadult male emitted a “shhek-shhek-shhek” wing-produced sonation very similar to A. rothschildi (Fig. 4C). However, it seems possible that the wing sonation is not produced with every wing beat, in this particular subadult at least. On more than one occasion documented in the video data, the subadult male leaves a perch and makes a short flight with no audible wing sound.

Visual components of Astrapia splendidissima—Splendid Astrapia

We identified one courtship behavior, the perch-hop, performed from the relatively open branches of an emergent tree on the periphery of a natural upland clearing. Because no females were documented, we treat these data as practice displays. It is unclear if the emergent tree was an established display site or merely opporunistic.

Perch-hop

This behavior is similar to displays of the other species in which the male performs a series of rapid flight-hops from branch to branch through the canopy of a tree (video vouchers: ML 462734, 462738, 462739 and 462748). It begins with the male on a perch in a slightly hunched body posture in which the feathers along the back are slightly expanded into a domed or hump-like shape similar to that of A. mayeri. Feathers around the face and neck are expanded, which creates a conspicuous feather “beard” below the lower mandible. Then he suddenly performs a flight-hop through the canopy of the tree and alights on another branch, if only very briefly, before jumping to other branches in a varied pattern around the crown of the tree with little vertical movement. The wings are spread occasionally to assist the jumping motion while the tail remains stiff and rudder-like. Sometimes the male returns to the perch were he began, while other times he does not, completing the varied pattern. At the conclusion of his display the male vocalizes by giving the harsh, froglike call (see below).

Acoustic components of Astrapia splendidissima—Splendid Astrapia

Like the previous species, the acoustic components of the A. splendidissima courtship phenotype are also relatively simple. We identified two sounds used in the context of courtship, one vocal and one non-vocal (audio vouchers: ML 163653, 163654 and video voucher ML 462766). Recordings of a male A. splendidissima from near Lake Habbema in Papua, Indonesia a reveal a three-note call that is a series of nasal frog-like notes. Compared with the others, this species appears to be much more vocal. The first two notes are very similar and the third is higher and with a distinct ringing or bell-like quality, “ger ger ti” or “jeer jeer ti” (Fig. 5D). A single recording of an individual from a population to the west of Lake Habbema shows a similar three-note pattern, but with a longer interval between each note and with all three notes being an identical buzzy, “jereet” (see ML 100385). As with the other species, the wings of A. splendidissima produce a non-vocal rattling sonation in flight (Fig. 4D).

Acoustic components of Astrapia stephaniae—Stephanie’s Astrapia

We identified two sounds used in the context of courtship, one vocal and one non-vocal (audio vouchers: ML 100595 and ML 100597). Recordings of multiple adult and subadult males interacting, including with females present, reveal that A. stephaniae gives a series of whistled wee-wheet notes somewhat similar to the more drawn out notes of A. mayeri (Fig. 5E). Another call is a shrill kri kri kri kri with a variable number of notes given in apparent display-like context with female plumage birds nearby (e.g., ML 100595). The wing-sounds of A. stephaniae are loud and conspicuous, like those of A. mayeri, with a distinct rattle-like quality (Fig. 4E).

Discussion

Relative to species in most other genera of birds-of-paradise, knowledge about the visual and acoustic components of the courtship phenotypes for species in the genus Astrapia has remained poorly known for a very long time. With just one exception (Healey, 1978), nothing about Astrapia courtship has been documented in the peer review literature since the first species was described in later part of the 18th century.

With regard to courtship behaviors in particular, previous knowledge was limited to perch-hopping in A. mayeri and A. stephaniae and an inverted display in A. rothschildi (Healey, 1978; Frith & Beehler, 1998). Here, we document seven distinct male courtship behaviors: (1) perch-hopping, (2) pivoting, (3) inverted tail-fanning, (4) nape-pecking, (5) post-copulatory tumbling, (6) upright sleeked posturing and (7) branch-sidling. We also document two female specific behaviors: display interaction and wing-fluttering. In addition, we document distinct species-specific vocalizations and wing-produced sonations for all five species.

Of the male behaviors, perch-hopping is the most broadly distributed among species with a version now known for all species but A. nigra. The details of the behavior among the species are quite similar. In all of them, the displaying bird moves quickly between multiple branches by hopping or making short flight-hops. In A. rothschildi, hopping between perches sometimes includes chasing females (or female plumaged individuals). Lack of data makes it unclear if chasing is component of perch-hopping in the other species. One significant difference in perch-hopping among species centers around body posture. In A. mayeri and A. splendidissima, the male adopts a distinctly hunched or “hunchbacked” posture while perch-hopping whereas A. rothschildi remains more upright in a posture that is not as obviously ritualized. We believe that a form of a perch-hop behavior is likely to exist within A. nigra, but confirmation will need to come from additional field observation.

A type of pivot display is now known from two species, A. rothschildi and A. mayeri. In both, it involves repeatedly moving in a ritualized fashion from side-to-side with feet more-or-less in a fixed position. The most distinctive feature of the A. rothschildi pivot is wing flicking whereas in A. mayeri, the most distinctive features are the very ritualized hunchbacked posture and the highly exaggerated swishing movement of the male’s long ribbon-like tail (Fig. 6). While purely speculative given the available data, we nevertheless surmise that a form of pivoting is likely to be present in the other species and will be discovered with more data. For example, a form of horizontal perch pivoting is broadly distributed among the species in the genus Parotia (Scholes, 2008c).

Of all the Astrapia courtship behaviors, the inverted tail-fan display of A. rothschildi and A. nigra is one of the most distinctive and specialized (Figs. 2, 3 and 8). Among birds-of-paradise, true inverted displays—i.e., those that involved hanging upside down underneath a horizontal branch—have only been discovered in three species from the Cicinnurus-Paradisea clade: the King Bird-of-Paradise (Cicinnurus regius), the Blue Bird-of-Paradise (Paradisaea rudolphi) and the Emperor Bird-of-Paradise (Paradisea guilielmi) (Frith & Beehler, 1998; Laman & Scholes, 2012). Several other species adopt head-down inverted postures from vertical or sloping branches (e.g., Seleucidis melanoleuca), but they do not hang fully inverted like the the three species mentioned above and the two species of Astrapia described here.

The overall similarity of the inverted displays of A. rothschildi and A. nigra is also quite striking (see Figs. 2, 3 and 8) and such detailed similarity in display behavior uncommon even among bird-of-paradise species in the same genus. For example, even though the ballerina dance display is shared by all the species in the genus Parotia, the details of the displays are nevertheless different among Parotia species with diagnosably distinct components in each (Scholes, 2008c). Because data documenting the inverted tail-fan displays of A. nigra are so few and likely incomplete, we feel confident that there are additional components still to be discovered, which will distinguish it from the inverted display of A. rothschildi. These differences are likely to be most noticeable in displays given to females, which is something that has not yet been documented in A. nigra.

Another notable feature of the inverted displays of A. rothschildi and A. nigra is that both species have highly iridescent abdominal plumage, which appears dark (almost black) when the birds are in normal uprights perching posture. However, when the abdominal feathers are oriented skyward during inverted display, they become highly visible (e.g., Fig. 2D). Interestingly, A. splendidissima also has highly iridescent green abdominal plumage, which begs the question about if it too has an undocumented inverted display behavior. According to the best-supported phylogeny available, A. nigra and A. splendidissima are sister species, which together are sister to a clade that includes A. rothschildi, A. mayeri and A. stephaniae with A. rothschildi as the basal member (Irestedt et al., 2009). This means that having green abdominal plumage is likely to be ancestral and the darker abdomens of A. mayeri and A. stephaniae are derived. It also implies that the inverted display behavior either evolved twice independently in A. nigra and A. rothschildi or that it, like green abdominal plumage, is also the plesiomorphic state and was present in the common ancestor to the extant Astrapia clade. If the second scenario is true, we can predict that A. splendidissima is likely have an inverted courtship display similar to those of A. rothschildi and A. nigra. Documenting the presence of, or confirming the absence of, an inverted display in A. splendidissima should be a high priority for future field observers.

We documented pre-copulatory nape-pecking behavior in two species, A. rothschildi and A. mayeri. However, we think this behavior is likely to also be present in some, if not all, of the other species and is likely to be discovered with more data. The reason we feel confident in predicting a form of nape-pecking is likely to be present in the other species is because nape-pecking during courtship is fairly broadly distributed throughout the birds-of-paradise. Outside of Astrapia, ritualized nape-pecking is also known from species in the genera Paradisea, Cicinnurus, and Parotia.

In contrast, we feel that the likelihood of a post-copulatory tumbling behavior in species other than A. rothschildi is much more uncertain. This behavior, with the male and female tumbling together toward the ground, is arguably the most unusual courtship behavior known from the genus and among the most unusual known from all the birds-of-paradise. In fact, nothing like it is known from any other species in the family.

Interestingly, anecdotal evidence from captive A. stephaniae suggests that something like the tumbling behavior described for A. rothschildi might be found in that species as well (Boehm, 1966; Frith & Beehler, 1998). Boehm (1966) writes, “The pre-copulation ceremonies of the Stephaniae are quite vigorous and brutal; they grasp one another with their talons, usually in the thighs, or in whatever manner chance may present.” Boehm then goes on to describe how when grasping one another they often fall to the ground, which is suggestive of the post-copulatory tumbling of A. rothschildi. However it is difficult to interpret the context of what Boehm describes given the captive circumstances and therefore solid conclusions about a post-copulatory tumbling behavior in A. stephaniae remain difficult to make. Future field observers should be on the lookout for post-copulatory behaviors in the other species.

The additinal two male display behaviors described and documented here, upright sleeked posture and branch-sidling, are both exclusive to A. mayeri. Both behaviors are only cursorily documented here and should be considered somewhat tentative until other details can be filled in. This does not mean we have doubts about their validity; rather, we think they are likely to have additional components or, perhaps they will be discovered to be parts of other, more complex, displays that were not evident from the limited data available here. Various forms of upright and or sleeked postures are known from species in a number of bird-of-paradise genera, including Cicinnurus, Ptiloris and Lophorina and so it is not too surprising that a similar behavior is found in the genus Astrapia. It is somewhat surprising that it was not found in other Astrapia species however. Future fieldworkers should be on the lookout for similar behavior in species other than A. mayeri. Branch-sidling behavior is not as widespread among the birds-of-paradise as are upright sleeked postures, however something like it is known from Parotia sefilata (Scholes, 2008c). In this particular case, however, we think the similarity is superficial and not evidence of any form of behavioral homology. Sidling behaviors should be looked for in the other Astrapia species, especially in A. stephaniae, which is the sister species to A. mayeri.

Because A. stephaniae is the only species with no behavioral components examined here, it is worth reviewing what is known about it from the literature. The most complete primary record of courtship in A. stephaniae comes from Healey (1978). Although lacking illustrations or other documentation of the specific display behaviors, this paper describes several variations on a perch hopping behavior, including one with a whistled vocalization followed by a hop from branch to branch within or among trees. It also describes a similar low- and high-intensity ritualized hopping between two perches within an established lek tree with other individuals, including female plumaged birds, in attendance. Healey describes the body being held vertical at the end of each perch hop and that the tail is swung forward under the perch (Healey, 1978; pg. 200). While the specifics are somewhat ambiguous, Healey also mentions observing a bird “spasmodically flicking” its wings open a few centimeters (Healey, 1978; pg. 199). Other behaviors and postures described by Healey include a hunched posture, expansion of the breast feathers and a raised “nasal tuft” (Healey, 1978; pg. 199). All of these behaviors (perch hopping, wing-flicking, hunched posture and ornamental plumage erection) are consistent with those described among the other species described in this study. Because of the obvious phenotypic similarities, the close phylogenetic relationship and documented hybridization (e.g., Fig. 5E) between A. mayeri and A. stephaniae, we believe that many elements of the A. stephaniae courtship phenotype are likely to be shared or similar to those of A. mayeri.

Although relatively better known than visual components and behaviors, the acoustic components of the Astrapia courtship phenotype have not been well documented before this study. While it remains true that Astrapia are indeed among the least vocal of the birds-of-paradise, each species nevertheless gives a distinct vocalization and all are now documented to give wing-produced sonations. The primary vocalization types within the genus fall into three categories: throaty growls (e.g., A. rothschildi), click or tick-like sounds (e.g., A. nigra and A. splendidissima) and whistled notes (e.g., A. mayeri and A. stephaniae). The “shek-shek” sounds produced by the wings in flight or during display when flicked (e.g., A. rothschildi) appear to be a plesiormorphic component of the Astrapia courtship phenotype. This observation is further supported by the fact that at least one of the two species in the genus Paradigalla, which is the sister group to the Astrapia, has also been reported to emit a ratting or rustling sound in flight (Frith & Beehler, 1998).

Our study uncovered two female behaviors, display interaction and wing-flutter, both of which were found only in A. rothschildi. Although not well described in the literature, wing-fluttering by females is a widespread female solicitation behavior that has been observed many birds-of-paradise and is quite widespread among birds in general. We therefore expect that females of most, if not all, Astrapia species engage in this behavior duiring courtship, especially prior to copulation.

Among the birds-of-paradise, fairly complex and highly interactive female courtship behaviors, like the one described here for A. rothschildi, have only been described in the primary literature for female Parotia carolae (Scholes, 2006). However, video recordings of other species and courtship descriptions in the secondary literature suggest the interactive female behaviors may not be entirely uncommon, but instead are just not well reported (e.g., female riflebirds, genus Ptiloris, lift their heads up and spread their wings during male displays and females of several other species are clearly active during courtship, not just passive observers). Because no courtship displays performed for females have been observed and/or documented in other species of Astrapia, it is currently unclear if other species have interactive female behaviors. However, given the fairly extensive form of this female behavior in A. rothschildi, we surmise that interactive female behaviors will be found in other species in time. Of particular interest is A. nigra since males of that species have a similar inverted tail-fanning display, which is the context in which female A. rothschildi were documented to give their interactive courtship behavior.

Conclusions

While comprehensive knowledge about courtship phenotypes for species in the genus Astrapia remains incomplete, our results nevertheless provide the most complete picture to date. We show that the courtship phenotypes of Astrapia species have a relatively rich suite of visual and acoustic components, including at least seven distinct male behaviors and two female specific behaviors. In addition, we found that all five species have distinct vocalizations and wing-produced sonations. Because our results come from analysis of fully vouchered and online accessible audiovisual media, they are readily available for reanalysis or reinterpretation. Our results provide a valuable framework for future research, including detailed comparative and evolutionary studies among these and other bird-of-paradise species.

We thank the curatorial, collections management and online database team from the Macaulay Library at the Cornell Lab of Ornithology for their roles in making the data in this study archived and accessible. We also thank the many field assistants, guides and field workers who made the original recordings possible. We thank Kimberly Bostwick, Gerald Borgia, Richard Prum and one anonymous reviewer for their comments and improvements to the manuscript.

Additional Information and Declarations

Competing Interests

Author Contributions

Data Availability

The authors declare there are no competing interests.

Edwin Scholes conceived and designed the experiments, performed the experiments, analyzed the data, contributed reagents/materials/analysis tools, wrote the paper, prepared figures and/or tables, reviewed drafts of the paper.

Julia M. Gillis conceived and designed the experiments, performed the experiments, analyzed the data, wrote the paper, prepared figures and/or tables, reviewed drafts of the paper.

Timothy G. Laman performed the experiments, contributed reagents/materials/analysis tools, reviewed drafts of the paper.

The following information was supplied regarding data availability:

Macaulay Library: https://macaulaylibrary.org/search?utf8=&taxon_id=11993792&taxon_rank_id=62&taxon=Astrapia.

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
