# Peer review of "Visual and acoustic components of courtship in the bird-of-paradise genus Astrapia (Aves: Paradisaeidae)"

_PeerJ, doi:10.7717/peerj.3987_

## Round 0.1 · original submission · Major Revisions

Please revise your manuscript according to the concerns of the reviewers. It is particularly important not to overstate the importance of your study (see comments by Reviewer 2).

·

Basic reporting

This paper involves description of the little known male display behavior r of the Bird of Paradise genus Astrapia from videos take by the authors. The detail of the description varies with species, the description of the Huon Astrapia has the greatest detail. The descriptions tend to be detailed and in some cases very limited for some species The authors cite the available literature. There are no hypotheses being tested as this is pure description and some speculation as to the behavior

Experimental design

This paper provides only descriptions of videos of male courtships and in one case a copulation with a female. The methods are adequate to to allow descriptions where relevant videos exist.

Validity of the findings

The findings appear valid. The first video of each behavior type and the behaviors described match. No statistics are presented due to small sample sizes.

Additional comments

The manner in which the male behaviors are described are very mechanical. While there is no doubt that males are following as sequence I would have found it beneficial to see these displays as interactive, and inclusive of the female's behavior. In some instance her behavior is noted, but more detail would be of interest and important in interpreting the actions on video. Even saying that she is off screen would be useful in the early stages. It would also be useful to have a short summary of the results of the behaviors being described at the beginning of the sequence. Courtships leading to copulations and those the do not are often somewhat different as are those in which young males may be involved.

Reviewer 2 ·

Basic reporting

This article meets the standards set out for basic reporting. However, I do think the introduction is exaggerated with respect to the importance and implications of this study.

Experimental design

This paper meets the standards set out for experimental design. However, there is no research question posed in this study. This work is simply a description of behaviors noted in videos present in a public data base. This paper does not present the results of a 'study' or 'experiment' or 'investigation' nor is there a 'research question'. It utilizes videos available in a public data base, and it describes behaviors that have not been described before.

Validity of the findings

It would appear that this study meets the standards for validity of the findings. The videos in the library at Cornell University are 'valid' and thus descriptions of behaviors observed in those videos are also therefore valid. But, again, this paper does not present the results of a study or research project.

Additional comments

I appreciate the value of the videos collected by one or more of the authors and it is very important that behaviors previously unknown are described. This 'ethogram' part of the current paper is fine. These behaviors and vocalizations are important to describe. But, that's all this paper is. It is a partial, incomplete, poorly documented ethogram of courtship behaviors based on video recordings. That's okay, but I beg to differ with the authors that these descriptions represent the courtship phenotype of this genus. I generally don't like that term courtship phenotype, but even if it is a good term the genus Astrapia does not have one courtship phenotype as suggested by the title and introduction. Each species would have its own courtship phenotype, and each species would exhibit considerable variation in that phenotype.
Specific comments:
1. The authors couch the introduction as it's important to study birds of paradise in order to understand how sexual selection promotes diversity. The notion that sexual selection promotes speciation is a hypothesis, not a well-accepted fact. It probably does, but it is certainly not widely accepted to do so. And, the data in this study have nothing to do with sexual selection. The data in this study are just descriptions of behaviors used in courtship. I think the introduction should be rewritten to emphasize how little is known about various species of birds of paradise and the importance of cataloging their behaviors so that comparative studies can be done. I think the authors are trying too hard to make this study seem relevant to sexual selection or behavioral ecology in general. This study has no relevance to sexual selection or behavioral ecology. It is relevant to the biology of birds of paradise and understanding the diversity of behaviors in that group. That's okay, but I think the authors should not try to overstate the importance of their data.
2. The authors should describe what is meant by 'core bird of paradise' or 'core bird of paradise genera'.
3. The authors conclude that there are considerably more behavioral components to the display of Astrapia species than previously known. Well, considering that there is not a single paper describing any courtship behavior in any species of Astrapia, isn't that obvious? What is the point of even saying this? If zero behaviors were previously described, and now there are 7 behaviors described, does that mean that there are many more behaviors than previously known? I'm not purposefully trying to be flippant, but my point highlights the point above that the authors are trying to make this partial ethogram into something it isn't. These behaviors are important to describe, but I encourage the authors to accept the extreme limitations of their data set and not to venture outside of that data set.
4. I would have appreciated a comparison of the display behaviors across the species. Perhaps there aren't enough videos of all species? But, I think the authors should try to compare the various display elements across the species that show them. Or, compare these behaviors to those seen in other genera, like Parotid that the senior author has studied extensively.

·

Basic reporting

All good.

Experimental design

No experiments were conducted.

Validity of the findings

The authors have been very responsible in restricting the interpretation of their data to generalizations supported by video observations. I would actually prefer more analysis and speculation in conclusions.

Additional comments

The authors present some extraordinary observations of the courtship display behaviors of Astrapia birds of paradise. These birds are incredibly poorly known, and found in some of the most remote habitats on earth. Practically ever page of the manuscript elicited at least one involuntary "Holy shit!" from this reviewer. The effort required to obtain these data are amazing, and the novelty is beyond doubt.
The real contributions of this manuscript is to further broaden our understanding of the diversity of the displays of birds, and birds of paradise in particular.

I think that the manuscript could be improved with a little bit of a change in narration style or perspective. The authors have presented this manuscript as an example of video "data mining". Although I applaud the use of video vouchers to document the behavior, it is not as if the authors are recommending this as a research method. Unlike genbank or other data bases, one can't legitimately say, "Gee, let's mine this video collection for amazing and unexpected data on bird behavior!" Incongruously, the author's don't reveal that Scholes and Laman have spent nearly a decade scouring the forests of New Guinea painstakingly collecting these kinds of video data!

I think attempting to present behavior as a more modern, data-mining pursuit actually distorts and harms our understanding of the results. For example, we are told that certain perches are "practice" perches, but we don't know anything about why that judgment was made. Because the results are not presented in terms of how many days were spent where and in what season of the year, etc., we don't know whether the lack of continuity in behavior was because the birds never returned to that perch again, or the researchers had to move off. That sort of thing happens a lot. The authors assume things from knowledge they have, but are not part of the 'data-mining' procedure, and this affects all of the results.

Do the authors really imagine that members of the public could "mine" this "publicly accessible" video collection, and do anything like this job of accurate job of analysis? Absolutely not! The difference between these alternatives, of course, is what is missing from the ms. Perhaps brief description of the field effort etc. that went into each batch of recordings, and some further details in supplemental files could address this problem accurately. But it is also one of intellectual stance that comes across throughout the ms.

Lastly, I think that the authors could do more to compare, or at least discuss, the elements of the behaviors that are shared or convergent with other birds of paradise. For example, the wing-shuffling mechanical sounds are shared with some species of Epimachus, no? What other birds of paradise perform these relatively rare inverted displays? In the realm of convergence, the Velvet Asity (Philepitta castanea) also does a sort of somersault display around the perch that could be mentioned. Is there anything about the display of A. mayeri that is specifically associated with its extraordinarily long, and derived white tail? Although these comments would be speculative, they would be very constructive. So the analysis at the end seems shorter than needed.


Miscellaneous comments (line numbers):

59- No "the" before Astrapia

94-98- Include number of males here.

160-161- What exactly is a pivot? Lots of side to side motions would not be called pivots, so this is unclear.

173- The phrase "a display that is called…" sound very Olympian, like the names of the displays are discovered. But the authors just made them up! Please use "that we call…"

200-01- a side to side rocking motion that makes the tail bob up and down? How is that? Can you help a little more?
247- "Interactive observation (female)" - Sounds like the name of an modern, avant garde painting. It's not even helpful because an interaction requires two individuals, and the sex of only one is identified. How about the straight up- "Male-female interactions" ?? Same goes for "Upright nape-peck (pre-copulatory)". (I can't wait to see that in a gallery in Soho!) Why not Pre-copulatory upright nape-peck ?
254- turn, not tern
279- I am sure that it is challenging to visualize, but I really recommend a sonogram of the shirk-shirk wing sound. Although it may look poorly, acoustical details can probably be extracted from it. I think that needs to be in, or its absence needs to be explained.
316, 318- Should be fig. 5, not 4.
602-3- I cannot visualize the bird and its parts in Fig 2C at all. What parts are wings or tails? Can the authors help with this- perhaps a little inset line drawing? Otherwise, this image is a blackhole.

---

## Round 0.2 · accepted · Accept

Thank you for the extensive revisions.